# Transdermal Delivery of Phloretin by Gallic Acid Microparticles

**DOI:** 10.3390/gels9030226

**Published:** 2023-03-15

**Authors:** Roberta Cassano, Federica Curcio, Roberta Sole, Sonia Trombino

**Affiliations:** Department of Pharmacy, Health and Nutritional Sciences, University of Calabria, 87036 Cosenza, Italy

**Keywords:** gallic acid, phloretin, microspheres, swelling, antioxidant, transdermal release

## Abstract

Exposure to ultraviolet (UV) radiation causes harmful effects on the skin, such as inflammatory states and photoaging, which depend strictly on the form, amount, and intensity of UV radiation and the type of individual exposed. Fortunately, the skin is endowed with a number of endogenous antioxidants and enzymes crucial in its response to UV radiation damage. However, the aging process and environmental stress can deprive the epidermis of its endogenous antioxidants. Therefore, natural exogenous antioxidants may be able to reduce the severity of UV-induced skin damage and aging. Several plant foods constitute a natural source of various antioxidants. These include gallic acid and phloretin, used in this work. Specifically, polymeric microspheres, useful for the delivery of phloretin, were made from gallic acid, a molecule that has a singular chemical structure with two different functional groups, carboxylic and hydroxyl, capable of providing polymerizable derivatives after esterification. Phloretin is a dihydrochalcone that possesses many biological and pharmacological properties, such as potent antioxidant activity in free radical removal, inhibition of lipid peroxidation, and antiproliferative effects. The obtained particles were characterized by Fourier transform infrared spectroscopy. Antioxidant activity, swelling behavior, phloretin loading efficiency, and transdermal release were also evaluated. The results obtained indicate that the micrometer-sized particles effectively swell, and release the phloretin encapsulated in them within 24 h, and possess antioxidant efficacy comparable to that of free phloretin solution. Therefore, such microspheres could be a viable strategy for the transdermal release of phloretin and subsequent protection from UV-induced skin damage.

## 1. Introduction

Ultraviolet (UV) radiation is responsible for damaging effects on the skin that occur either acutely or on a delayed basis. An example of an acute effect is inflammation caused by cytokines that results in erythema or “sunburn,” which is characterized by the reddening of the skin caused by the sun. In addition, such radiation increases cellular levels of reactive oxygen species and is responsible for phenomena such as oxidative stress [1], a pathological condition that occurs when an abnormally high amount of free radicals is produced in a living organism, exerting a damaging action on DNA, cells, and tissues, particularly the skin, in which there is an increase in fibroblasts, mast cells, macrophages, and T lymphocytes, and, thus, the development of an inflammatory process. All this leads not only to the establishment of photoaging processes, but also to a wide variety of chronic-degenerative diseases [2,3,4,5]. To perform its protective function, the skin has developed endogenous antioxidant and cytoprotective defense systems and produces specific detoxifying enzymes against reactive oxygen species [6]. Unfortunately, however, the aging process and environmental pollution can reduce the production of these endogenous defense systems. Therefore, natural antioxidants can be an important resource against UV-induced skin damage [7]. In fact, humans are unable to synthesize these compounds de novo, and several plant foods constitute the natural source of various antioxidants [8], such as carotenoids, polyphenols, etc. [9]. These include gallic acid and phloretin used here by us.

Gallic acid (GA), or 3,4,5-trihydroxybenzoic acid, is a carboxylic acid of the phenolic type found in many types of plants and particularly in grapes, tea, hops, and oak bark. It has anti-inflammatory, antioxidant [10,11], antifungal, antibacterial, antiallergic [12], and anticarcinogenic [13,14] properties.

Phloretin is a member of the flavonoid class dihydrochalcones and is distributed mainly in green apples and strawberries (Figure 1) [15]. The compound has antioxidant, anti-inflammatory [16,17], antidiabetic [18,19], anticancer, and antimicrobial properties [20]. However, phloretin’s poor water solubility reduces its absorption and, consequently, bioavailability, limiting its application in the treatment of various diseases and administration via conventional drug delivery systems [21].

To facilitate the topical administration of phloretin, it is possible to use formulations able to improve the dissolution of the drugs, increasing their stability and bioavailability [22,23,24,25,26,27].

For this purpose, polymeric microspheres based on gallic acid have been made here for the transdermal administration of phloretin (Figure 2).

The decision to use gallic acid for particles formation is due to its interesting chemical structure. In fact, it has two types of functional groups, carboxyl and hydroxyl, which are amenable to derivatization. Moreover, gallic acid, which is present in the microsphere structure, can both act synergistically with the phloretin trapped inside and preserve it from oxidation processes. In fact, phloretin, in addition to being poorly soluble in water, is an unstable molecule that can degrade especially upon prolonged exposure to aqueous solutions. The microspheres, based on gallic acid, were obtained by reverse-phase emulsion radical polymerization reaction and characterized by transformed infrared spectroscopy. Their antioxidant efficacy was tested on rat liver microsomal membranes. Swelling behavior, phloretin loading efficiency, and transdermal delivery were also evaluated. All the results obtained indicated the possibility of using these microparticles for the transdermal delivery of phloretin and subsequent protection from UV-induced skin damage.

## 2. Results and Discussion

### 2.1. Esterification of the Gallic Acid with Methyl Alcohol

This reaction, which represents the first step to obtaining the esterified gallic acid, required the use of DCC and DMAP. Methanol was used as both a solvent and a reagent. (Figure 1). The reaction was carried out in anhydrous conditions to avoid the degradation of DCC by water. The obtained mixture was deprived of solvent under reduced pressure, and raw residue was recrystallized from methanol. This allowed the methyl gallate to be obtained, which was characterized using FT-IR, GC/MS, and ^1^H-NMR. FT-IR (KBr) ν (cm^−1^): 3445 (–OH), 3329 (–OH), 3034 (–CH), 2945 (–CH), 2928 (–CH), 1760 (–C=O), 1627 (–C=C), 1244 (–CO). MZ: 56 (100%), 184 (2%). ^1^H-NMR (CD_3_OD) δ (ppm): 7.250 (2H, d), 3.829 (3H, s). The yield was 98%.

### 2.2. Transesterification of Methyl Gallate with Allyl Alcohol

The methyl gallate (1) was subjected to trans-esterification with allyl alcohol. The potassium *tert*-butoxide removes a proton from the hydroxyl group of allyl alcohol, leading to the formation of an alcoholate, an extremely reactive nucleophile, that by attacking the carbonyl group of 1, promotes the release of the methoxy group and leads to the formation of the trans-esterified gallic acid (Figure 2). The obtained product was hydrolyzed with acidic water and extracted with chloroform. The combined organic phases, dried under reduced pressure, gave a product (2) that was thoroughly characterized by FT-IR spectrophotometry, GC-MS, and ^1^H-NMR. FT-IR (KBr) ν (cm^−1^): 3446 (–OH), 3333 (–OH), 3034 (–CH), 1729 (–C=O), 1626 (–C=C), 1261 (–CO), 990 (–CH), 892 (–CH). *m*/*z*: 56 (100%), 168 (7%). ^1^H-NMR (CD_3_OD) δ (ppm): 6.853 (2H, d), 5.41 (1H, ddt), 5.05 (1H, dd), 5.01 (1H, dd), 4.66 (2H, d). The yield was 79%.

### 2.3. Preparation of the Microspheres Based on Allyl Gallate

The microspheres were obtained by reversed-phase emulsion radical polymerization reaction. In particular, the aqueous solution of a monomer, which represents the dispersed phase, was added to an excess of organic solvents, immiscible with water, as the dispersing phase. Thus, under the action, small droplets of the dispersed phase formed and assumed a spherical shape. The reaction was carried out in a cylindrical reactor in which was added trans-esterified gallic acid previously solubilized in distilled water (dispersed phase). The dispersing phase, based on chloroform and n-hexane, had previously been introduced into the reactor and kept under stirring conditions. The formation of the microspheres was initiated by the addition of ammonium persulfate, radical initiator, and the comonomer N, N′-methylene-bis-acrylamide. More importantly, Span85 and Tween85 were also involved in this reaction. In fact, these surfactants contribute to the formation of the spherical structure of the particles. The TMEDA (N,N,N′,N′-tetramethyl-ethylenediamine) allows the process of the decomposition of the radical initiator to be accelerated. The reaction was carried out under constant mechanical stirring at 40 °C for three hours. The obtained microspheres were washed with three solvents in succession (isopropanol, ethanol, and acetone) and then were dried.

### 2.4. Microspheres Characterization

The microspheres were characterized by means of light scattering and FT-IR, which confirmed the disappearance of the typical bands of the allyl group at 990 cm^−1^ and 892 cm^−1^ and the appearance of a new band of carbonyl stretching at 1634 cm^−1^. The light scattering analysis showed that the obtained microspheres had a good polydispersity (0.095 ± 0.001) and that the average particle diameter was equal to 1.5 ± 0.13 µm. Morphological analysis, by using scanning electron microscopy (SEM), showed that the particles, with micrometric dimensions, appeared to be grouped in part in large clusters (Figure 3).

### 2.5. Swelling Degree Evaluation 

The microspheres swelling degree (α%) was evaluated at appropriate time intervals (1 h, 2 h, 3 h, 4 h, 6 h, 12 h, 24 h), using Franz cells and a solution of water/ethanol 8/2. The swelling degree was calculated using the following equation:$$\alpha \%=\frac{Ws-Wd}{Wd}\times 100$$

*Ws* and *Wd* represent the weights of swollen and dried microspheres, respectively [28]. Each experiment was carried out in triplicate. The results, reported in Figure 4, show the achievement of a good swelling degree after 2 h. This value remained almost constant up to 24 h, validating the possible use of microspheres for phloretin transdermal delivery.

### 2.6. Phloretin Loading Efficiency

The microspheres were impregnated with a solution of water/ethanol 8/2 of phloretin and left under stirring at 37 °C for 72 h. Subsequently, the solution was filtered and analyzed by UV-VIS (λ = 288 nm, ε = 3201.4 mol^−1^∙dm^3^∙cm^−1^). This allowed the loading efficiency (*LE%*) to be calculated through the following equation: $$LE\%=\frac{Ci-Cf}{Cf}\times 100$$

*Ci* represents the initial drug concentration in solution, while *Cf* is the drug concentration in solution after the loading study. *LE%* was 67%.

### 2.7. In Vitro Skin Permeation Studies

The ability of the microspheres to release phloretin in contact with cellulose acetate membranes or the rabbit skin was evaluated after 1 h, 2 h, 3 h, 4 h, 6 h, 12 h, and 24 h using Franz cells. The obtained data showed that the microspheres released the phloretin efficiently in contact with the skin, and the cumulative percentage of active substance released in 24 h was comparable with the release of phloretin solution (Figure 5). In particular, about 90% of the loaded phloretin was released within 24 h.

The GC/MS analysis demonstrated that the microspheres protected phloretin from the degradation process. In fact, phloretin was almost integrally released after approximately 30 min from the acid gallic-based microspheres (Figure 6A). On the contrary, phloretin in solution, released under the same conditions, suffered degradation (Figure 6B).

### 2.8. Antioxidant Activity Evaluation

The ability of the microspheres, loaded and not loaded with phloretin, to inhibit lipid peroxidation, induced by a source of free radicals, such as *tert*-butyl hydroperoxide (*tert*-BOOH), which endogenously produces alkoxyl radicals by Fenton reactions, was examined in rat liver microsomal membranes of Wistar rats (250–300 g) (Charles River Laboratories, Lecco, Italy) over 120 min of incubation [28]. The antioxidant activity of the prepared materials was time-dependent and, as can be seen from Figure 7, was preserved in time. The obtained data indicated that the microspheres containing phloretin possessed a higher antioxidant efficacy with respect to the unloaded ones.

## 3. Conclusions

In this work, phloretin, a natural antioxidant substance with many therapeutic properties, was incorporated into gallic acid microspheres. Tests were performed that showed that the swelling of the particles increases with time, facilitating the release of phloretin from the matrix. Preliminary studies designed to evaluate the release behavior of phloretin have shown that the particles effectively release phloretin upon skin contact, with performance comparable to that of free drug solution. In addition, the microspheres were able to protect phloretin from degradation, as confirmed by mass spectrometry analysis. The antioxidant activity of loaded and unloaded gallic acid microspheres, tested in rat liver microsomes, was found to be time-dependent and higher for microspheres containing phloretin than for those that did not contain it. All the results obtained indicate that the microspheres, based on gallic acid, could be applied as a transdermal carrier of phloretin for the prevention and treatment of UV radiation-induced skin damage.

## 4. Materials and Methods

### 4.1. Reagents

Acetone, hydrochloric acid, chloroform, diethyl ether, ethanol, isopropanol, methanol, n-hexane, tetrahydrofuran (THF), allyl alcohol, and sodium sulfate were purchased from Carlo Erba Reagents (Milan, Italy). Gallic acid (MW = 170.12), phloretin (MW = 274.27), dicyclohexylcarbodiimide (DCC), N,N-dimethylaminopyridine (DMAP), potassium *tert*-butoxide, N,N-dimethylacrylamide (DMAA), ammonium persulfate (NH_4_)_2_S_2_O_8_, N,N′-methylene-bis-acrylamide, sorbitan trioleate (Span85), polyoxymethylene sorbitan trioleate (Tween85), N,N,N′,N′-tetramethyl-ethylenediamine (TMEDA), *tert*-butylhydroperoxide (*t*-BOOH), trichloroacetic acid (TCA), 2-thiobarbituric acid (TBA), and butylated hydroxytoluene (BHT) were Purchased from Sigma-Aldrich (Sigma Chemical Co, St. Louis, MO, USA).

### 4.2. Instruments

The IR spectra were performed using a spectrometer FT-IR Perkin Elmer 1720. ^1^H-NMR spectra were obtained using a spectrometer Bruker VM30; the chemical shifts were expressed as δ and referring to the solvent. The structures of the obtained compounds were confirmed by mass spectrometry using a Hewlett Packard instrument GM-MS Hewlett Packard 5972 (Analytical Instrument Management, Littleton, CO 80127, USA). The UV-VIS spectra were realized by means of UV-530 spectrophotometer JASCO. Dimensional analysis of the microspheres prepared were carried out by means of light scattering using a Brookhaven 90 Plus Particle Size Analyzer. Scanning electron microscopy (SEM) photographs of microspheres were obtained with a JEOL JSMT 300 A.

### 4.3. Animals

The animal study protocol was approved by the Italian Ministry of Health (Rome, Italy) (protocol code 700A2N.6TI, date of approval: March 2018). 

### 4.4. Esterification of Gallic Acid with Methyl Alcohol (1)

In a three-neck flask fitted with a reflux condenser and magnetic stirring, fully flamed, and maintained under nitrogen atmosphere, gallic acid (2 g, 1.17 × 10^−2^ mol) was dissolved in dry methanol (13 mL) [29]. The reaction was left under reflux and magnetic stirring at 35 °C for 30 min. After that, DCC (2.41 g, 1.17 × 10^−2^ mol) and DMAP (1.43 g, 1.17 × 10^−2^ mol) were added, and the system was stirred for 30 min. The reaction, kept under reflux for 12 h at 50 °C and magnetic stirring, was monitored using thin layer chromatography (TLC/silica gel, eluent phase chloroform-methanol mixture 7:3). At the end, the reaction mixture was hydrolyzed and extracted with chloroform (3 × 10 mL). The orange organic phase was dried under vacuum. Next, the obtained solid was purified by recrystallization from methanol. The white precipitate (1) was recovered by filtration and characterized through FT-IR spectrophotometry, GC-MS, and ^1^H-NMR. 

### 4.5. Transesterification of Methyl Gallate with Allyl Alcohol (2)

The second step involved the trans-esterification of the compound (1) with allyl alcohol. The reaction was carried out according to a procedure reported in the literature [30]. In a three-neck flask, fitted with a reflux condenser and magnetic stirring, fully flamed, and maintained under nitrogen atmosphere, allyl alcohol (0.33 mL, 8.06 × 10^−3^ mol) was dissolved in dry THF (30 mL). The solution was heated to 80 °C and stirred. The temperature was then lowered to 70 °C and potassium *tert*-butoxide (0.98 g, 8.06 × 10^−3^ moles) was added after 30 min. The methyl gallate (1.49 g, 8.06 × 10^−3^ moles) was added. The reaction was monitored using thin layer chromatography (TLC/silica gel, eluent chloroform-methanol mixture in the ratio 8:2). Then the reaction mixture was hydrolyzed with acid water and extracted with chloroform. The orange organic phase was dried under vacuum. The white product (2) was characterized through FT-IR spectrophotometry, GC-MS, and ^1^H-NMR.

### 4.6. Preparation of Microspheres Based on Allyl Gallate

Microspheres, based on allyl gallate, were obtained by radical polymerization reaction in reverse phase emulsion according to the procedure reported in literature [30,31]. Briefly, a cylindrical glass reactor of 100–150 mL, equipped with mechanical stirrer, dripping funnel, and screw cap with puncture-proof rubber septum, was flamed in a nitrogen flow; after cooling, it was immersed in a bath thermostatically controlled at 40 °C. Next, n-hexane (20 mL) and chloroform (18 mL) were introduced into the reactor. After 30 min of N_2_ bubbling, this mixture was treated with distilled water containing allyl gallate (0.1 g, 4.7 × 10^−4^ mol), the co-monomer methylene bisacrylamide (MBA, 0.037 g, 2.38 × 10^−4^ mol), and ammonium persulfate 800 mg (3.5 × 10^−3^ moles) as radical initiator. The mixture, under stirring at 1000 rpm, was treated with 150 μL of sorbitan trioleate (Span85), then after 10 min, with 150 μL of polyoxymethylene sorbitan trioleate (Tween85), and after a further 10 min, with 150 μL of N,N,N′,N′-tetramethyl-ethylenediamine (TMEDA). This mixture was left under stirring conditions for another 3 hours [26]. The obtained microspheres were filtered, washed several times with 100 mL of isopropanol, 100 mL of ethanol, and 100 mL of acetone to remove all traces of free acrylic moieties, co-monomer, and initiator, and dried under vacuum at 40° C overnight. Their characterization was carried out by light scattering, FT-IR spectrometry, and SEM.

### 4.7. Phloretin Loading Efficiency

Microspheres (0.49g) were soaked for 3 days at room temperature, under magnetic stirring, in a phloretin solution. The phloretin (0.01 g, 3.5 × 10^−5^ moles) was solubilized in 10 mL of distilled water/ethanol 8/2 solution. The amount of solubilized drug was chosen to have a drug loading of 20% (*w*/*w*). After 3 days, the microspheres were filtered and dried, at reduced pressure, in the presence of P_2_O_5_ at constant weight. 

### 4.8. Size Distribution Analysis

The size of the particles was determined by dynamic light scattering (DLS) using a 90 Plus Particle Size Analyzer (Brookhaven Instruments Corporation, New York, NY, USA) at 25 °C by measuring the autocorrelation function at 90° scattering angle. Cells were filled with 100 µL of sample solution and diluted to 4 mL with filtered (0.22 µm) water. The polydispersity index (PI), indicating the measure of the distribution of particle population [28], was also determined. Six separate measurements were made to derive the average. Data were fitted by the method of inverse “Laplace transformation” and Contin.

### 4.9. Swelling Degree Evaluation

The swelling behavior of the microspheres was investigated to check their hydrophilic affinity. The study was realized through Franz cells using a solution water/ethanol 8/2. Predetermined aliquots of dried microspheres (0.018 g) were placed in the Franz cells with a solution of water/ethanol. At predetermined time intervals (1, 2, 3, 4, 6, 12, 24 h), microspheres were deprived of excess water, weighed, and finally their swelling degree was calculated. Each experiment was carried out in triplicate (*n* = 3).

### 4.10. In Vitro Skin Permeation Studies

Skin permeation tests were performed using Franz Diffusion cells (*n* = 3) both with cellulose acetate membranes and rabbit ear skin (New Zealand rabbits 2.9–3.1 kg; provided by local butcher) for 24 h with the aim to compare eventual components of skin interferences. The apparatus was maintained at 37.0 °C to mimic physiological conditions. Receptor chambers (6.0 mL) were filled with NaCl 0.9% solution and kept under stirring conditions. Free phloretin was used as control. At specific time intervals (1, 2, 3, 4, 6, 12, and 24 h) aliquots (7 mL) of each sample were withdrawn from receptor chambers and replaced with fresh release medium. Samples were analyzed through UV-Vis spectrophotometry (288 nm) and GC-MS. In particular, after release, all the solutions withdrawn intended to be characterized by GC/MS were dewatered and then subjected to analysis after solubilization in ethanol.

### 4.11. Antioxidant Activity Evaluation

First, 1 mL of microsomal suspension, reacted with microspheres, was mixed with 3 mL 0.5% TCA and 0.5 mL of TBA solution (two parts 0.4% TBA in 0.2 M HCl and one part distilled water), and 0.07 mL of 0.2% BHT in 95% ethanol. Samples were then incubated in a thermostatic bath at 90 °C for 45 min. After incubation, the TBA–MDA complex was extracted with 3 mL of isobutyl alcohol. The absorbances of the extracts were measured by UV spectrophotometry at λ = 535 nm, and the results were expressed as mmol per mg of protein [32].

### 4.12. Statistical Analysis 

All quantitative data were expressed as means ± standard deviations. Differences between means were analyzed for statistical significance using the Student’s *t*-test. In this study, *p*-values less than 0.05 were considered statistically significant.

## Data Availability

Not applicable.

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
