# Peer review of "Transdermal Delivery of Phloretin by Gallic Acid Microparticles"

_gels, 2023, doi:10.3390/gels9030226_

Round 1
Reviewer 1 Report
Dear author
Thank you for the presented work.
Introduction needs more elaboration and include more references to support hypothesis.
Scheme 2, characters that are not clear appear under the arrow, check it please
Figure 6 caption is not clear, rephrase please
What are the evidence that support employing gallic acid as phenolic acid. Citation required for similar work presented in literature.
Line 57 page 2 Rephrase the sentence please, provide number or illustrate chemistry related solubilty.
use uppercase for litre abbreviation please.
Line 252 grammar/language
Line 294 grammar/ language
Figure 4 is not clear. You claim in text, drug was eluted after 24 h while 30% was released. It may appear that only 30% was released after 1 h and this was measured. So either 70% is trapped or was not loaded. Also it does not look like cumulative release. Confusing representation
Please revise captions and improve presentation of the results.
I was not able to detect NMR chart for the newly synthesized compounds. If not very clear it can be supplementary. Also IR charts.
Line 329 In vitro skin permeability is not accurate. NO skin used, only cellulose paper. This will only provide idea of amount diffused. An alternative experiment may be added using skin or similar artificial membranes. So if you have ethical approval where did you use the animal experiment? that was not clear. So you have materials, instrumentation, animals, then esterfication, transesterfication, preparation of microsphere... etc.
Animal skin can be used to detect diffusion, permeability and skin deposition.
Statistics need revising as the differences and similarity are not clear.
Best wishes.
Author Response
REVIEWER 1
Thank you for the presented work.
- Introduction needs more elaboration and include more references to support hypothesis.
The introduction has been implemented and more references have been added
- Scheme 2, characters that are not clear appear under the arrow, check it please
Schema 2 has been improved
- Figure 6 caption is not clear, rephrase please
Figure 6 caption was corrected
4)What are the evidence that support employing gallic acid as phenolic acid. Citation required for similar work presented in literature.
Liu and collaboartors in a recent review, reported that: “Gallic acid (GA), 3,4,5-trihydroxybenzoic acid, is one of the most abundant phenolic acids in fruits and medicinal plants. It is a colorless or slightly yellow crystalline compound with promising therapeutic and industrial applications”.
This review is cited among our references
Liu J., Yong H., Liu Y., Bai R. Recent advances in the preparation, structural characteristics, biological
- Line 57 page 2 Rephrase the sentence please, provide number or illustrate chemistry related solubilty.
This sentence:” The limited solubility of these substances in water and lipids results in poor ab-sorption and bioavailability, limiting their use in traditional drug delivery systems” has been replaced with “phloretin's poor water solubility reduces its absorption and, consequently, bioavaila-bility, limiting its application in the treatment of various diseases and its administra-tion by conventional drug delivery systems”
- Use uppercase for litre abbreviation please.
As suggeted by reviewer we used uppercase for litre abbreviation
- Line 252 grammar/language
We do not understand what the reviewer wants to indicate. Also, we removed much of this paragraph by mentioning only the protocol number.
- Line 294 grammar/ language
The sentence “Then, the required amount of n-hexane (20 ml) and chloroform (18 ml), constituting the dispersant phase, was introduced into the reactor” was changed in “Then, n-hexane (20 ml) and chloroform (18 ml) were introduced into the reactor”.
- Figure 4 is not clear. You claim in text, drug was eluted after 24 h while 30% was released. It may appear that only 30% was released after 1 h and this was measured. So either 70% is trapped or was not loaded. Also it does not look like cumulative release. Confusing representation
Please revise captions and improve presentation of the results.
As suggested we revised the Figure 4 now Figure 5 and its caption inserting the phloretin cumulative release percentage that was equal to ~90%.
I was not able to detect NMR chart for the newly synthesized compounds. If not very clear it can be supplementary. Also IR charts.
Unfortunately, we are unable to recover the original GC/MS, FT-IR, and NMR spectra due to issues with a previous brownout that caused unplugged instruments to fail and lose all data. For the same reason we do not currently have the possibility to carry out an FT-IR spectrum of the microspheres. We were only able to do light scattering which allowed us to confirm the stability of the microspheres. Furthermore, we had not asked ourselves the problem since the numerical values as we reported them in the manuscript were almost always sufficient.
-Line 329 In vitro skin permeability is not accurate. NO skin used, only cellulose paper. This will only provide idea of amount diffused. An alternative experiment may be added using skin or similar artificial membranes. So if you have ethical approval where did you use the animal experiment? that was not clear. So you have materials, instrumentation, animals, then esterfication, transesterfication, preparation of microsphere... etc.
-Animal skin can be used to detect diffusion, permeability and skin deposition.
As reviewer suggested we laso performed in vitro skin test using animal skin.
Statistics need revising as the differences and similarity are not clear.
We introduced a new paragraph as follows:
All quantitative data were expressed as means ± standard deviations. Differences between means were analysed for statistical significance using the Student’s t test. P-values less than 0.05 were considered statistically significant.

Reviewer 2 Report
The study "TRANSEDERMAL DELIVERY OF PHLORETIN BY GALLIC ACID MICROPARTICLES" is interesting, however, it is poorly designed. There are significant gaps in the methodologies and results, as well as a lack of FTIR, DLS, and animal data. Here are a few major suggestions to help improve the article.
1. The abstract should include clear mathematical facts reported by the authors.
2. It is recommended to include plant sources of gallic acid.
3. Because the authors noted the toxicity of GA and Phloretin, it is best to discuss them in depth for a better understanding. Furthermore, the authors noted that both compounds are weakly water and lipid soluble; do the molecules belong to BCS class 4? The authors should clarify their statement.
4. Rewrite lines 74-86 (Gallic acid was chosen because …… efficiency and transdermal release.) for better understanding.
5. The authors should investigate the hydrolytic degradation of the microspheres after soaking them for three days.
6. It would be preferable if the authors included Phloretin entrapment efficiency data
7. What is the rationale for using water/ethanol 8/2 instead of any physiological buffer condition?
8. The protocol of the in vitro drug release research should be included in the method section, or the heading should be changed accurately in both the results and method sections.
9. Provide a reference to section 4.11.
10. Check the scheme 1 “50癈” and correct it
11. For section 2.2, the author should provide the GC-MS, 1H-NMR, and FT-IR spectra for better understanding.
12. Section 2.3 should be rewritten for clarity.
13. The author should include FT-IR data for the characterization of microspheres . Correct the particle size format 1,5±0,13 µm to 1.5±0.13 µm. Include the SD for polydispersity as well. Figure 3's resolution could be improved.
14. What is the reason for the high swelling at 2 hours but poor swelling over the next 24 hours?
15. Provide a scatter plot of phloretin release at pH 6.8 and pH 7.4 for a better understanding. The bar graph shows that there has been no change in the release profile. Figure 4 claims two distinct pH conditions, although it is the study of Phloretin in microsphere and solution. Provide the data's standard deviation
16. Figure 5 represents a different sale bar that does not give any information about the outcome of degradation. Provide high-resolution GC/MS data to Figure 5.
17. The author should disclose the findings of in vitro skin permeation research
18. ention the grouping and dose of the animal research in section 2.8. For data clarity, provide pathophysiological proof of non-loaded and loaded microspheres of rat liver microsomal membranes.
Author Response
REVIEWER 2
The study "TRANSEDERMAL DELIVERY OF PHLORETIN BY GALLIC ACID MICROPARTICLES" is interesting, however, it is poorly designed. There are significant gaps in the methodologies and results, as well as a lack of FTIR, DLS, and animal data. Here are a few major suggestions to help improve the article.
- The abstract should include clear mathematical facts reported by the authors.
As suggested the abstract was revised.
- It is recommended to include plant sources of gallic acid.
In the text this sentence is reported: “Gallic acid (GA), known as or 3,4,5-trihydroxybenzoic acid, is a carboxylic acid of the phenolic type found in many types of plants and particularly in grapes, tea, hops, and oak bark”.
Gallic acid used for our purposes was of Sigma Aldrich with purity ³ 98 %.
- Because the authors noted the toxicity of GA and Phloretin, it is best to discuss them in depth for a better understanding.
The term toxicity, improperly used, does not refer to gallic acid or phloretin but to the ability, in general, of nanoformulations to increase stability, bioavailability and reduce drug toxicity. Therefore, in order not to create confusion for the reader, we have decided to drop the term "toxicity".
- Furthermore, the authors noted that both compounds are weakly water and lipid soluble; do the molecules belong to BCS class 4? The authors should clarify their statement.
Our statements are based on literature data indicating that the water solubility of phloretin is 0.023±0.001 g/L. With regard to gallic acid its solubility is of 1.19 g/100 mL. Then both substances aren’t totally unsoluble in water.
- Rewrite lines 74-86 (Gallic acid was chosen because …… efficiency and transdermal release.) for better understanding.
As suggested by reviewer, we rewrote the sentence and the new is:
“We decided to use gallic acid, for particles formation, thanks to its interesting chemical structure. In fact, it has two types of functional groups, carboxyl and hydroxyl, which are susceptible to derivatization”.
- The authors should investigate the hydrolytic degradation of the microspheres after soaking them for three days.
As requested, we studied the hydrolytic degradation of the microspheres after immersing them for three days and did not notice any degradative effects.
- It would be preferable if the authors included Phloretin entrapment efficiency data
The Phloretin entrapment efficiency, that we call "phloretin loading" has been indicated as LE % and it is equal to 67%.
- What is the rationale for using water/ethanol 8/2 instead of any physiological buffer condition?
We used ethanol 20% to increase phloretin solubility and to promote its release.
- The protocol of the in vitro drug release research should be included in the method section, or the heading should be changed accurately in both the results and method sections.
We included the protocol in the method section.
- Provide a reference to section 4.11.
The follwing reference was inserted: 27.32. Trombino, S.; Poerio, T.; Curcio, F.; Piacentini, E.; Cassano, R. Production of α-Tocopherol–Chitosan Nanopar-ticles by Membrane Emulsification. Molecules 2022, 27, 2319-2330.
- Check the scheme 1 “50癈” and correct it
We do not understand what the reviewer wants to indicate.
- For section 2.2, the author should provide the GC-MS, 1H-NMR, and FT-IR spectra for better understanding.
Unfortunately, we are unable to recover the original GC/MS, FT-IR, and NMR spectra due to issues with a previous brownout that caused unplugged instruments to fail and lose all data. For the same reason we do not currently have the possibility to carry out an FT-IR spectrum of the microspheres. We were only able to do light scattering which allowed us to confirm the stability of the microspheres. Furthermore, we had not asked ourselves the problem since the numerical values as we reported them in the manuscript were almost always sufficient.
- Section 2.3 should be rewritten for clarity.
As requested, section 3.2 has been rewritten.
- The author should include FT-IR data for the characterization of microspheres.
Unfortunately, we are unable to recover the original GC/MS, FT-IR, and NMR spectra due to issues with a previous brownout that caused unplugged instruments to fail and lose all data. For the same reason we do not currently have the possibility to carry out an FT-IR spectrum of the microspheres. We were only able to do light scattering which allowed us to confirm the stability of the microspheres. Furthermore, we had not asked ourselves the problem since the numerical values as we reported them in the manuscript were almost always sufficient.
Correct the particle size format 1,5±0,13 µm to 1.5±0.13 µm.
The particle size format has been corrected.
Include the SD for polydispersity as well.
The SD for polydispersity has been included.
Figure 3's resolution could be improved.
The resolution has been improved
- What is the reason for the high swelling at 2 hours but poor swelling over the next 24 hours?
The nature of the material with which the particles have been prepared allows water to enter by diffusion and interaction with the polymer matrix. The interactions that are established between the particle matrix and the water mean that the degree of swelling increases in the first two hours faster and then remains almost constant over 24 hours even to an extent dependent on the chemical nature of the matrix and the interaction between matrix and solvent.
- Provide a scatter plot of phloretin release at pH 6.8 and pH 7.4 for a better understanding. The bar graph shows that there has been no change in the release profile. Figure 4 claims two distinct pH conditions, although it is the study of Phloretin in microsphere and solution. Provide the data's standard deviation
By mistake we misreported the caption information which has been corrected.
- Figure 5 represents a different sale bar that does not give any information about the outcome of degradation. Provide high-resolution GC/MS data to Figure 5.
Unfortunately we do not have the possibility to recover the original GC/MS, FT-IR and NMR spectra due to problems with a voltage sag ago which caused all data to be lost. Furthermore, we had not asked ourselves the problem given that the numerical values as we reported them in the manuscript were almost always sufficient.
- The author should disclose the findings of in vitro skin permeation research
In vitro skin permeation studies has been reported in 2.7 section
- Mention the grouping and dose of the animal research in section 2.8.
We inserted in the section 2.8 the type of rats used: of Wistar rats (250-300 g) (Charles Rives Laboratories Lecco Italy).
For data clarity, provide pathophysiological proof of non-loaded and loaded microspheres of rat liver microsomal membranes.
It is impossible to make a pathophysiological determination of the unloaded and loaded microspheres with rat liver microsomal membranes since the latter are used as a suspension and not as a tissue.

Reviewer 3 Report
1- The abstract must be rewritten. Some important results of the research must be added to the abstract. And also, the abstract is too short. Add some parts of the introduction, methodology, and results.
2- Is Figure 1 necessary to show? Mainly with the picture of the fruits.
3- Page 2, lines 74-84 need some references for the gallic acid properties and applications.
4- Page 3, line 91: this sentence has grammatical errors. And also put a dot for the last sentence of the current line.
5- Scheme 1. Write down the name of each compound under the chemical structure. And what is that symbol after 50 and 70, under the arrow in schemes 1 and 2?
6- As shown in Figure 3, the shape of the microspheres is not spherical and there is a diversity in the size of the particles. This can be affected by the drug release phenomenon. Why the size of the particles is different and how can the authors solve this problem?
7- The swelling ratio measurement is not presented in detail. Also, instead of using the table, draw the data in a graph with error bars.
8- Line 170: I cannot see Ci and Co in the equation. How did the authors calculate these parameters? Instead, please explain what is Ai, and Af?
9- Increase the resolution and quality in Figure 4.
10- In Figure 5, what are A and B? Mention them in the caption.
11- The authors are supposed to use the following references to improve the manuscript:
Sabbagh, F., Muhamad, I. I., Nazari, Z., Mobini, P., & Khatir, N. M. (2018). Investigation of acyclovir-loaded, acrylamide-based hydrogels for potential use as vaginal ring. Materials Today Communications, 16, 274-280.
Sabbagh, F., & Kim, B. S. (2022). Recent advances in polymeric transdermal drug delivery systems. Journal of controlled release, 341, 132-146.
Author Response
REVIEWER 3
- The abstract must be rewritten. Some important results of the research must be added to the abstract. And also, the abstract is too short. Add some parts of the introduction, methodology, and results.
As suggested the abstract was totally revised.
- Is Figure 1 necessary to show? Mainly with the picture of the fruits.
We revised the picture.
3- Page 2, lines 74-84 need some references for the gallic acid properties and applications.
From references 9 to 13 the properties and values of gallic acid are reported.
4- Page 3, line 91: this sentence has grammatical errors. And also put a dot for the last sentence of the current line.
We corrected the grammatical errors.
5- Scheme 1. Write down the name of each compound under the chemical structure. And what is that symbol after 50 and 70, under the arrow in schemes 1 and 2?
As reviewer requested the name of compounds were added under the chemical structures.
6- As shown in Figure 3, the shape of the microspheres is not spherical and there is a diversity in the size of the particles. This can be affected by the drug release phenomenon. Why the size of the particles is different and how can the authors solve this problem?
The microspheres have a spherical shape and high dimensional uniformity as confirmed by light scattering. However, gradually they tend to interact giving rise to aggregates of variable shape and size
7- The swelling ratio measurement is not presented in detail. Also, instead of using the table, draw the data in a graph with error bars.
As suggested we reported the swelling ratio measurement in agraph with error bar sas follows:
8- Line 170: I cannot see Ci and Co in the equation. How did the authors calculate these parameters? Instead, please explain what is Ai, and Af? (cambiare equazione)
Due to a typing error we have reported another equation which we have corrected.
9- Increase the resolution and quality in Figure 4.
We changed the figure 4 increasing resolution.
10- In Figure 5, what are A and B? Mention them in the caption.
This following caption was inserted:” Figure 5: Evaluation, through GC/MS analysis, of phloretin released, after approximately 30 min, from acid gallic-based micro-spheres (figure 5A) and from a solution (figure 5B) in the same conditions”
11- The authors are supposed to use the following references to improve the manuscript:
Sabbagh, F., Muhamad, I. I., Nazari, Z., Mobini, P., & Khatir, N. M. (2018). Investigation of acyclovir-loaded, acrylamide-based hydrogels for potential use as vaginal ring. Materials Today Communications, 16, 274-280.
Sabbagh, F., & Kim, B. S. (2022). Recent advances in polymeric transdermal drug delivery systems. Journal of controlled release, 341, 132-146.
We inserted the indicated references.

Round 2
Reviewer 1 Report
Dear author
Thank you for addressing the comments indicated to you.
Reviewer 2 Report
Dear Authors
Please go through the English grammar and spelling one more time
Reviewer 3 Report
The authors have developed the quality of the manuscript and it can be publishable in the journal.